# IoT-Enabled Smart Cities: A Review of Concepts, Frameworks and Key Technologies

**Pierfrancesco Bellini** , **Paolo Nesi \*** and **Gianni Pantaleo**

Distributed Systems and Internet Technology Lab (DISIT), University of Florence, 50139 Florence, Italy; pierfrancesco.bellini@unifi.it (P.B.); gianni.pantaleo@unifi.it (G.P.)
* Correspondence: paolo.nesi@unifi.it

**Abstract:** In recent years, smart cities have been significantly developed and have greatly expanded their potential. In fact, novel advancements to the Internet of things (IoT) have paved the way for new possibilities, representing a set of key enabling technologies for smart cities and allowing the production and automation of innovative services and advanced applications for the different city stakeholders. This paper presents a review of the research literature on IoT-enabled smart cities, with the aim of highlighting the main trends and open challenges of adopting IoT technologies for the development of sustainable and efficient smart cities. This work first provides a survey on the key technologies proposed in the literature for the implementation of IoT frameworks, and then a review of the main smart city approaches and frameworks, based on classification into eight domains, which extends the traditional six domain classification that is typically adopted in most of the related works.

**Keywords:** smart cities; internet of things; big data

## 1. Introduction

The increasing development and dissemination of Internet of things (IoT) and Internet of everything (IoE) technologies represent an important enabler in the current smart cities landscape, leading the smart city paradigm to the big data scale. In fact, as noted in [1], a technical report by Ericson estimated that 29 billion devices would be connected by 2022. According to the report presented by Statista Research in 2019, it has been estimated that the total number of connected IoT devices worldwide will increase to about 75 billion by 2025 [2], creating a potential IoT economic impact that could reach USD 11 trillion per year by 2025 [3]. These numbers are revealing that IoT is going to be one of the highest value disruptive technologies, opening new frontiers, possibilities and challenges in the production of smart services and applications. The importance of IoT is more and more closely connected to the evolution of smart cities, for which IoT represents one of the key drivers for smarter innovation and sustainable development. Smart cities are complex socio-technical infrastructures, composed of human actors (different stakeholders and users, such as citizens, city operators, administrative institutions, public and private companies, etc.) and digital devices (e.g.: city sensors and actuators exploited in many domains, such as mobility and transportation, environment, energy, healthcare, governance, industry 4.0 etc.; smart home and smart buildings devices; and personal devices, such as smartphones). This complexity is reflected in the large variety of heterogeneous approaches, contexts, application domains and technological solutions that have been proposed in the literature for the realization and management of smart cities. The main goal of implementing and integrating IoT solutions is to allow smart cities to advance further, providing new capabilities and features while significantly reducing human intervention [4]. In addition, it is also important to focus on the social challenges addressed and the societal benefits achieved by the adoption of these technologies, for instance assessing how they can contribute to the Sustainable Development Goals (SDGs) developed by United Nations within the context

of the 2030 Agenda [5]. Some of the main technical challenges for modern IoT-enabled smart cities are represented by the requirements of supporting a multitude of different data providers, dealing with many different protocols and data formats, as well as ensuring interoperability and scalability and supporting the sharing of components. This is crucial in order to avoid the implementation of redundant solutions for data ingestion, storage and analysis, thus lowering operative costs and enhancing the city's sustainability. The motivation and contributions of the present paper are the following:

1.  Provide an updated and comprehensive (although not exhaustive) overview of research literature about smart city domains, solutions and frameworks, as well as about key IoT technologies and applications integrated in smart city components.
2.  Provide insights into recent trends, open technical and social challenges (also assessing the contribution of IoT–smart city technologies and domains toward the SDGs) and future directions to be addressed in the implementation of IoT in smart cities.

In this paper, an effort has been made to integrate IoT and smart city solutions following a classification approach that focuses on the identification of eight applicable domains: governance; living and infrastructures; mobility and transportation; economy; industry and production; energy; environment; and healthcare. The rest of the paper is organized as follows. In Section 2, a review of the main technologies employed in IoT frameworks is presented. Section 3 provides a survey focused on IoT-enabled smart city domains and components. Section 4 illustrates some recent trends and open challenges, and also assesses the societal challenges involved and investigates potential future directions for IoT-powered smart cities. Section 5 is left for the conclusions.

## 2. IoT Technologies and Architecture

The Internet of things has introduced an important paradigm innovation in the communication among digital devices, as well as in the way that they connect with the physical environment [1] and interact with human users. IoT architectures integrate and unify all steps of data sensing/actuating from/to devices, transmitting/receiving messages, data storage, processing, analysis and final exploitation through the use of cloud, fog and edge computation, services and applications. Many different technologies are involved in the development of IoT frameworks. However, general functional architectures have been proposed in the literature that are based on a simplified version of the Open Systems Interconnection (OSI) model, but with different approaches. Some works [1,6] define a typical IoT framework composed of three layers (from the lowest physical level to the higher levels of abstraction): the perception/sensing layer; the network layer; and the application layer. In other works [7–12] the former approach is expanded into five layers of architecture, including: the perception/sensing layer; the transportation/network layer; the middleware/processing layer; the application layer; and the business layer. In Figure 1, the three-layer and five-layer IoT architectures are depicted.

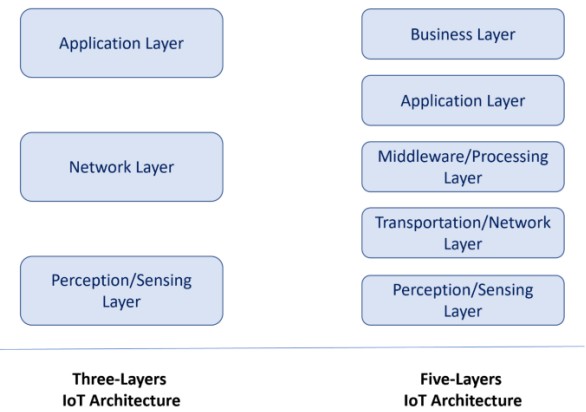

**Figure 1.** The most common IoT architectures (three-layer and five-layer) retrieved from the literature.

Different computing paradigms are defined depending on which level of the stack is being considered, from bottom to top: edge computing; fog computing; and cloud computing [13]. In the following, a description of the different layers is provided in terms of the functionalities and involved technologies. The five-layer architecture was considered since it can be viewed as a more detailed version of the three-layer architecture.

### 2.1. Perception/Sensing Layer

The perception or sensing layer is related to the physical level that is made up of devices, such as sensors and actuators, which interact among themselves and with the physical world by sending and receiving data, exploiting wireless networks [8]. In this context, sensors, actuators and mobile device technologies are involved. There is a wide range of commercial devices that measure many kinds of physical quantities and variables, i.e., sensors for measuring temperature, humidity, pressure, distance and geospatial coordinates, speed, acceleration, voltage, weight, pollutant particles, luminance level, biometrical signals, etc. Actuators are employed to control/move other devices or systems physically or virtually, and they are typically classified into electrical, pneumatic and hydraulic categories [14]. Several software tools and solutions are used to deploy low-level IoT applications. Those described in the following are based on visual programming languages (VPLs): NetLab is an open source environment for developing embedded systems; Ardublock is an open source visual block programming tool for Arduino systems; and Scratch is a visual programming tool, developed at MIT, for IoT code generation and communication with Arduino-based IoT products [15].

### 2.2. Transportation/Network Layer

The transportation or network layer provides the function of data routing and transmission. Network gateways/brokers can be used as mediators for integrating different IoT nodes, allowing them to transmit and receive data to and from different sensors for M2M connectivity. Regarding data transmission, different technologies and protocols are used. Proximity communication protocols include Bluetooth, radio-frequency identification (RFID) tag technology and near-field communication (NFC). Larger coverage networks exploit wireless technologies, such as Wi-Fi, Zigbee, long range wide-area network (LoRaWAN), Sigfox and 5G.

Bluetooth and Bluetooth Low Energy [16] are low-power wireless communication technologies designed for personal area networks (PANs), which are suitable for low-bandwidth data transmission among mobile devices over a short range of up to 10 m. They are usually employed in smart contexts for connecting personal devices.

RFID exploits radio frequencies for data communication and is usually used to uniquely identify objects, people, vehicles, etc. Communication is established between a reader device and a tag device, which can be passive or active. NFC [17] is similar to RFID, but it has an even shorter range for communication (up to few centimeters). Unlike RFID, NFC does not implement a reader/tag hierarchy and can be used for two-way communication. It is typically used for mobile payments and access control operations.

Wi-Fi is based on the IEEE 802.11 standard and uses wireless frequencies (operating at 2.4, 5 and 60 GHz bands) to provide high-speed internet connectivity (from 1 Mb/s to 6.75 Gb/s) within a limited distance (up to 100 m) [18]. It is designed for wireless local area networks (WLAN).

WiMAX (Worldwide Interoperability for Microwave Access) is based on the IEEE 802.16 standard, operating at 2–66 GHz bands and providing data rates from 1.5 Mb/s to 1 Gb/s [18]. It can support broadband wireless access for up to 50 km for fixed stations and between 5–15 km for mobile stations [19].

Zigbee is based on the IEEE 802.15.4 standard. It is a low-power and low cost protocol for wireless sensor networks (WSNs). It usually supports star, tree and mesh network topologies. The Zigbee protocol operates at 2.4 GHz and provides a data rate of about 250 kbit/s [20]. The data transmission range is similar to that of Wi-Fi (from 10 to 100 m).

LoRaWAN is a low-power wide-area network (LPWAN) that is capable of transmitting over long ranges (about 10 km) and also supports multitenancy and multi-domain networks [14]. It consists of several gateways that can be added when the network size increases and a higher capacity is required.

Narrowband IoT (NB-IoT) is an LPWAN protocol that operates on LTE licensed frequency bands, with a data rate of about 200 Kb/s [3]. The bandwidth of NB-IoT is about 180 KHz, and this protocol allows the connection of the order of hundreds of thousands of devices [21]. The NB-IoT technology also optimizes energy consumption, providing power control and power saving modalities [22].

LTE-M is another LPWAN protocol, operating at a 1.4 MHz bandwidth [23]. It was introduced with the aim of supporting a massive number of devices, providing 10 year battery life support [24] and supporting data rates of up to 1 Mbit/s, which is enough to satisfy the higher data rate requirements for devices such as video cameras and wearables [25]. The features supported in LTE-M include handover management, extended discontinuous reception and the suspension/resumption of radio resource control connection.

Z-Wave is a low-power protocol typically used in wireless home area networks, operating at 868 MHz and 900 MHz frequencies [7]. Z-Wave devices cannot directly connect to the Internet or other mobile devices. Therefore, the Z-Wave network is based on the use of a controller, which acts as a gateway, in order to manage all connected devices and allow them to interact with other mobile devices via the Internet or local networks [26].

Sigfox is a narrowband communication system for transmitting data over long ranges (up to 40 km [27]). Although it employs narrowband signals, Sigfox can be suitable for many kinds of applications, such as geolocation services and control messages.

Finally, 5G is the fifth generation of cellular networks. It is a very low-latency (less than 1 ms) and high-bandwidth (10 Gb/s) protocol [28], making it a strong enabler for smart cities and allowing the interconnectivity of a large number of IoT devices, as well as being suitable for real-time processing.

The last part of the network layer stack is in charge of suitably formatting data for their presentation. The client/server architecture, represented by the subscription mechanism between IoT devices and brokers, presents different modalities for sending/receiving messages/data, i.e., push and pull. Typically, pull protocols are REST call, web services, FTP and HTTP/HTTPS. On the other hand, the most common push protocols to receive data via data-driven subscriptions are: WebSocket (WS), Constrained Application Protocol (CoAP), Message Queue Telemetry Transport (MQTT), Advanced Message Queuing Protocol (AMQP) and FIWARE NGSI and NGSI-V2 [29]. Furthermore, in order to allow data exchange among multiple devices and applications, distributed publish–subscribe messaging systems should be supported (such as Apache Kafka, RabbitMQ, Orion Broker of FIWARE) for handling multiple data streams in an efficient and scalable way [13]. In this context, in order to increase fault tolerance when handling data that should not be lost in case of failure, the data persistence of queues is an important feature (e.g., Apache Kafka).

*2.3. Middleware/Processing Layer*

The middleware or processing layer can serve many different functionalities. For instance, it can act as a data aggregator module, since data may be collected from heterogeneous devices with different protocols that may not have been originally designed to communicate and interact with each other. Therefore, the middleware layer has to enable interoperability among connected devices, performing the necessary programming and/or model abstractions. Interoperability should be ensured at different levels:

- Technical level [30], in order to efficiently achieve and ensure end-to-end connectivity among devices, gateways, brokers, servers, etc.;
- Syntactical level, for managing the variety of protocols and formats;
- Semantic level, for exploiting Semantic Web technologies, such as XML, RDF, OWL Ontology and linked data (LD), to achieve unambiguous data representation and data semantic enrichment, thus improving the expressiveness level of the system [31].

IoT middleware is also typically devoted to providing scalability and reliability, allowing the system to handle a growing number of IoT connections and communication loads at the big data scale as well as supplying stable and fault-tolerant services. Moreover, in order to enable data processing over IoT, database-level data persistence is needed to integrate the data coming with different protocols in a shared model, and this is performed in the middleware layer. Therefore, middleware includes the different data storage modalities and eventually covers other functionalities, such as context identification, information extraction and the reconciliation of collected data. For example, FIWARE provides different generic enablers (Cygnus, QuantumLeap, STH-Comet) to store data coming from the Orion Context Broker.

### 2.4. Application Layer

The application layer provides the output formats, applications and services requested by the final users. Usually, works in the literature that are based on the three-layer IoT architecture include the messaging protocols management in this layer, as described in the network layer at the end of Section 2.2.

The growing dissemination of IoT devices and systems has led to the increasing adoption of event-driven applications (exploiting push protocols). This is a significant paradigm shift from the older generation of smart city applications that relied on vertical applications and were often based on extract, transform, load/extract, load, transform (ETL/ELT) processes and languages, which usually only support pull protocols. On the other hand, different frameworks and ecosystems are used to design and implement event-driven IoT application, for example VPL tools [32]. One of the most used is Node-RED, which is based on the Node.js engine and allows the creation of application flows in a graphic environment through the composition of visual nodes or blocks [33].

### 2.5. Business Layer

The business layer was introduced to classify all operations and front-end tools that consume data from the application layer for producing advanced big data analytics and visualization services, with the goal of building business models, supporting decision-making processes and performing simulations and what-if analysis. This can be achieved by implementing, for instance, predictive models based on machine learning, deep learning and artificial intelligence (AI) techniques [34], as well as advanced and interactive visual analysis tools [35]. Moreover, the business layer includes all operations performed by system administrators, which are needed to assess, control and maintain the overall functionality of the platform/framework.

IoT security aspects deserve a separate and specific discussion. In fact, security requirements and related issues have to be addressed along the full IoT stack, including all functional architecture layers, from the authentication of personal devices with IoT brokers and ensuring secure communication and secure encrypted storage for private data to the authentication mechanisms for accessing and using IoT applications, data analytics and data visualization tools. All of these requirements have to follow and cope with strict regulations, such as the General Data Protection Regulation (GDPR) standard [36].

## 3. Review of IoT-Enabled Smart City Components and Solutions

There is a wide range of research literature regarding IoT application in smart city contexts. A search of the Web of Science (WoS) database for papers containing the keywords "smart city" OR "smart cities" AND "IoT" OR "Internet of things" in their topic (i.e., the union of the "title", "abstract" and "keywords" search fields) resulted in a total of 5285 articles, published from 2010 to 2021, which is a very large number of resources to be extensively and systematically reviewed, or even to be filtered in a supervised way to select and consider the most relevant papers. In order to provide an overview of the growing interest around these topics, the temporal evolution of the above-mentioned papers (grouped by year of publication) is reported in Figure 2. It is to be noted that the apparent

decrease in the number of published papers over the last year (depicted with a dotted line in Figure 2) may be due to the fact that the count for 2021 was incomplete (since the search was performed in October 2021).

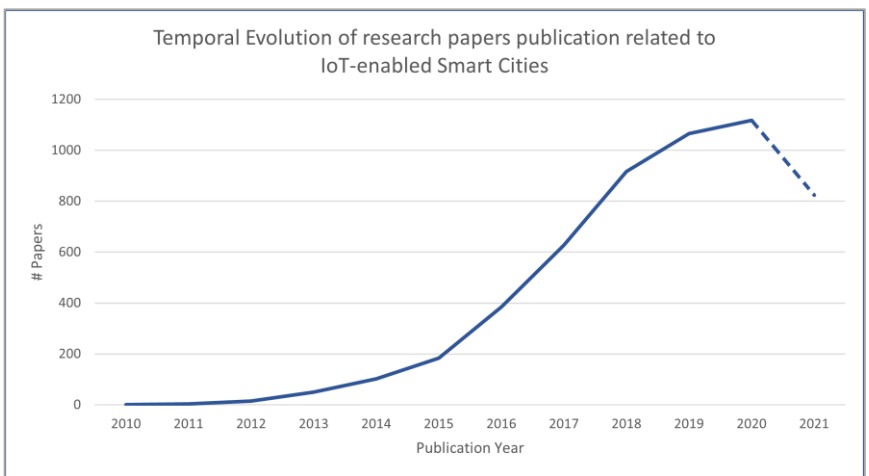

**Figure 2.** The temporal evolution of research papers related to IoT-enabled smart cities, as retrieved from the WoS database (collected in October 2021).

On the other hand, many different approaches have been proposed in the literature for classifying smart city frameworks and solutions in a variety of application domains. For this reason, we focused on reviews and surveys as a starting point for our study. Therefore, the literature research was conducted adopting the following criteria:

1. The WoS database was used for searching for reviews and survey articles containing the keywords "smart city" OR "smart cities" AND "IoT" OR "Internet of things" in at least one of the following fields: title; abstract; and paper keywords. Subsequently, a supervised overview and filter was performed in order to assure that each paper topic actually fit the subject of this review;
2. Recent literature was the main object of the present review, i.e., papers published from 2018 to the present (2021) were selected from the initial search;
3. Papers from Q1 and Q2 journals (as ranked in the SCImago index) were given priority over those from Q3 and Q4.

Following these criteria, a total of 52 surveys and reviews on IoT-enabled smart cities were considered as the baseline for the survey presented in this paper. It is to be noted that most of the 52 surveys reviewed often aimed to review one or some specific IoT–smart city domains and use cases without providing a more general or comprehensive overview, which is one of the aims of this paper. In addition, in our opinion, the few general surveys that were retrieved lacked in addressing the relationships between each smart city domain (with related sub-domains and scenarios) with the related IoT technologies that have been employed in each specific context.

In order to describe the wide landscape that we found by reviewing the selected literature in the most comprehensive way, the following eight domains were identified (as depicted in Figure 3), which are typically used to classify smart city components and application areas: governance; living and infrastructures; mobility and transportation; economy; industry and production; energy; environment; and healthcare. This approach extends the six-domain classification presented in [1,37]. The classification proposed in this paper is not meant to be exhaustive and, in some cases, these domains may not necessarily be orthogonal as they may overlap in several contexts and applications.

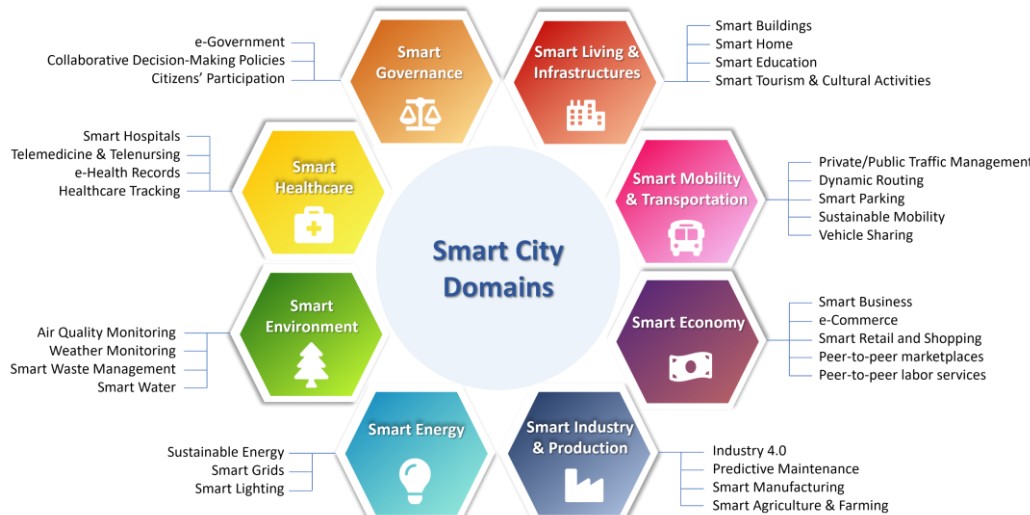

**Figure 3.** The classification of smart city domains with related components and application areas.

In the next subsections, each of the above identified smart city domains is discussed in terms of general scenarios, features and services provided, the IoT technologies employed, the frameworks or solutions adopted and real-world case studies of smart cities implementing solutions for that specific domain.

### 3.1. Smart Governance

Smart governance deals with the adoption of ICT into city governance practices in order to improve the decision-making process and speed up bureaucratic and administrative procedures through a smarter collaboration among different stakeholders and social actors [38], including public administrations, city officers, private companies and citizens. This can be successfully accomplished by providing innovative city services, dedicated channels and network integration for citizens. For instance, citizens can be engaged in participating with city governance activities and decisional processes through ICT-based tools and social media [39] as evidence of the mobile crowdsourcing paradigm [40], according to which citizens can act as "users as sensors" with their smartphones and mobile devices, participating as individuals and in groups in the acquisition process of data of interest for smart communities.

IoT technologies are transforming traditional city governance transactions and processes into smart government resources on the basis of the different participating actors, i.e., government-to-citizen (G2C), government-to-business (G2B) and government-to-government (G2G) [41,42]:

- Government-to-citizen (G2C) refers to the set of software solutions (typically web and mobile based) that support the relationships between public administrations and citizens, such as public administration web portals and/or mobile applications and social media channels employed for communication and interaction between local governments and citizens. In addition, IoT technologies, such as RFID and biometric sensors, are widely and increasingly adopted in electronic ID cards and mobile devices for identity recognition, electronic authentication and signature, according to different governmental standards, such as the European Community's electronic IDentification, Authentication and trust Services (eIDAS) [43]. These features are typically required to access services that are provided by public administrations and consult citizens' personal data related to public services, etc., thus simplifying a lot of the communication and interaction between governmental authorities and citizens;
- Government-to-business (G2B) regards the interactions between public administrations and businesses companies. In this model, e-procurement solutions are adopted, i.e., digital tools (mainly via the web), through which local governments publish ten-

ders, projects, competitions, facilities for the purchase/sale of goods and other general services for and from private companies. IoT technologies are widely adopted in G2B activities, facilitating and enhancing the relationship between local governments and companies that provide public and private services to citizens. For instance, transportation companies use location-based sensors (usually exploiting GPS technology) and services, sharing them with local administrations and allowing the easier and more efficient urban planning for mobility and transportation [44]. In addition, it includes similar aspects involved in many other domains that provide public and fundamental services, such as waste management, water, energy, etc. To this end, the use of cloud computing is generally adopted to store and share data and results among the different stakeholders (city operators, companies and citizens);

- Government-to-government (G2G) is related to the software solutions that aim to improve communications between the different public administration entities and groups, thereby speeding up all processes that require the interaction of these actors. This implies the use of IoT technologies for data collection, storage and sharing, which exploits, for instance, cloud computing and web/mobile-based services. A report of the European JRC also observed that governments may benefit from the combination of various data sources (e.g., from IoT and the web) with suitable analytical techniques (including AI-based techniques) to better identify and design specific administrative policies [45].

In Barcelona, ICT has represented a key driver for the evolution of the city model, based on flexible and efficient e-government initiatives and services that aim to make the city more innovative, inclusive and self-sufficient [46]. In the early 2000s, the city of Barcelona started implementing the *22@Barcelona* Project to enhance the technological, socio-economical and sustainable impact of the region in order to improve citizens' quality of life (QoL) [47]. In Amsterdam, a consortium consisting of the municipalities, research centers and private companies of the metropolitan area developed and launched the Amsterdam Smart City Platform (ASCP) in 2014: an online board where the aforementioned stakeholders can discuss urban issues, propose solutions and foster city innovation [48]. The local government in Rio de Janeiro has promoted city data sharing initiatives, hosting them in the so-called Intelligent Operations Centre (IOC) premises with the aim of increasing the efficiency of city services [49]. Moreover, the Rio Agora social platform has been implemented to allow citizens to propose and discuss public policies with municipal authorities [3]. Singapore has adopted several smart governance projects through the Smart Nation and Digital Governance Group (SNDGG) [50], including a digital identity system for Singapore residents that allows them to perform easier transactions with public administration and the Core Operations Development and eXchange (CODEX), which is a digital platform delivering smart services to citizens. In Toronto, the Sidewalk Labs smart city project was proposed. The project promotes data sharing standards and data governance models to profitably exploit the data gathered from citizens [51] through the urban data trust model. This allows for the collection, management and aggregation of urban non-personal data, de-identified data and personal information (according to data privacy regulations and laws) and then extracts value from them [52]. Songdo in South Korea is considered one of most advanced large-scale greenfield-based smart city projects in the world [53]. In fact, Songdo is defined as a ubiquitous U-city, in which data are continuously collected through a capillary network of sensors and equipment. Data are shared and interconnected with information from public institutions, such as the Incheon City Traffic Information Center, the Korean Meteorological Administration, the Institute of Health and Environment, Police Agency, etc. Data are analyzed by the U-Integrated Operation Center and provided to citizens through media broadcasts and control servers in order to help them to find the necessary information [54].

*3.2. Smart Living and Infrastructures*

The smart living domain includes all components related to developing smarter city infrastructures (e.g., smart homes, smart buildings, etc.) as well the management and improvement of public services, such as cultural activities, tourism and education, which are involved in enhancing the general quality of life of citizens:

- Smart Buildings: IoT allows the rapidly growing implementation of many kinds of facilities for smart buildings, for instance, air conditioning management, rainwater drainage, security systems for managing authenticated access to buildings, video surveillance and human activity monitoring [55], alerts for events such as fires and gas leaks, tools for monitoring the structural integrity of buildings [56], etc. Many different IoT technologies are involved in the living and infrastructures domain, depending on the specific use case or scenario. As for smart buildings, IoT integration with Building Information Modeling (BIM) tools provides a high-fidelity representation of buildings and spatial properties as a set of virtual assets [57], i.e., a digital twin of the building;
- Smart Homes: In these environments, different kinds of sensors, actuators and personal devices are connected through wireless networks and are often powered by human–machine interfaces that are based on artificial intelligence to provide smart and automated services for the users, with the goal of assisting them in daily tasks, such as lighting control, surveillance, managing home appliances and home resources, energy consumption, etc. [58]. In addition, smart home applications can be useful to detect and track the actions of the house's residents in order to monitor their health conditions [12], thereby especially helping the elderly and disabled people. Several kinds of sensors are applied to smart home and indoor sensing contexts. For instance, microelectromechanical systems (MEMS) are employed for the detection of gas leaks [59]. Devices based on triboelectric nanogenerators (TENG) are used for smart windows [60] and smart indoor lighting systems [61]. Video cameras and Closed-circuit television (CCTV) systems are used for smart surveillance. Digital humidity and temperature (DHT) sensors are largely adopted in fire alerting systems [62]. The most recent generation of domestic appliances and entertainment devices is often powered by AI and assistive services, so that they can interact while interconnected through wireless networks (the most used network communication protocols are Bluetooth, Zigbee, infrared and Wi-Fi [63]), thus providing users with a better, more efficient and enjoyable home life and experience;
- Smart Living Services: IoT devices have a large application in a variety of areas and activities that contribute toward improving the general quality of life for smart citizens. Cultural activities, for instance, smart tourism management, are taking advantage of exploiting mobile applications, GIS-aware and location-based services, multimedia streams, virtual and augmented reality and social media to manage and offer a better experience for tourism stakeholders [64]. Examples of these applications include tourist experience enhancement, destination competitiveness and sustainability improvement by tracking users' flows and behaviors [65]. Education is experiencing an increasing decentralization process with the inclusion of ICT and IoT elements, and this allows the production of new education services that can enhance interaction in remote and real-world learning activities [66].

Numerous smart living and infrastructures applications can be found in real-world cases. In [67], it is reported that 840 million units of smart home products were supplied in China in 2019. Los Angeles has implemented smart strategies for tourism management. Tourist traffic is measured through sensors embedded in the pavements and this information is used to adjust site visitor lights [68]. In Dubai, tourists can discover places of interest and events by exploiting NFC tags from their personal devices. There are many dedicated mobile apps that take advantage of cloud computing and allow visitors to connect seamlessly, without the need to download the app [69].

### 3.3. Smart Mobility and Transportation

The smart mobility and transportation concept implies the shift from traditional transportation systems to Mobility-as-a-Service (MaaS), where a smart IoT infrastructure connects different actors (citizens, public administrations, private companies) and entities (vehicles, personal devices, city sensors, actuators, etc.) [70]. IoT and intelligent transportation systems (ITSs) [71] allow the provision of smart applications and services to manage, for instance, private and public traffic flows, dynamic traffic routing, smart parking, vehicle sharing and sustainable mobility, connected driving, etc. Intelligent traffic solutions often rely on the application of predictive models for early warnings, accident prevention and real-time traffic congestion management.

In this context, many IoT network technologies have been proposed and used. City sensors and actuators (e.g., for the management of traffic lights, digital signage, road barriers, etc.), location-based GPS services and mobile-to-mobile communication are the basis for the implementation of vehicle-to-vehicle communication (V2V) and vehicle-to-infrastructure communication (V2I). Additionally, 5G networks and LTE-based systems have been employed for vehicle-to-everything (V2X) services [72]. Vehicular ad hoc networks (VANETs) [73] have been designed to handle a large number of nodes (including vehicles, roadside units (RSU) and on-board units (OBU) [74]) and present a good adaptability to the frequent topology changes and the data exchange rate [75]. The adoption of ITSs is at the basis of smart parking and car sharing [76] services. The most disseminated devices are RFID tags, infrared and ultrasonic sensors, smartphone-based sensors, video cameras for smart parking [77] and on-board diagnostic systems (OBD tools) [78].

In Berlin, the smart mobility service Jelbi was introduced in 2019, based on a mobile application that is connected to all services provided locally (e.g., car-sharing companies such as MILES and DB Flinkster and e-vehicle hire companies such as TIER), to foster the passage toward a more sustainable mobility [79]. In Florence, the Snap4City platform has been developed in the context of the Sii-Mobility project for sustainable mobility, providing a flexible IoT smart city platform and several applications to manage heterogeneous and complex urban mobility scenarios by integrating city sensors/actuators and IoT/IoE [34]. Atlanta has powered its transportation infrastructure with many smart technologies. For instance, a system of adaptive traffic signals that adjust to traffic conditions in real time are installed on the North Avenue Smart Corridor (a critical longitudinal communication roadway) [3]. London has a cycle rental system, currently offering more than 11,500 bikes and over 750 docking stations. As for car-sharing solutions, London has six active operators in the city, covering the whole Greater London area [80]. In addition, London's public transportation system provides the use of Oyster cards, which are smart electronic cards that exploit RFID technology and can be used for paying and buying electronic tickets for public transportation, thereby reducing queues and eventually tracking users' flows. In Santander, smart mobility services are applied to the management of public transportation, outdoor parking management and traffic routing [81]. The city of Bilbao has enhanced the sustainability of local mobility by deploying a network of e-bikes with 40 pick-up points and e-buses that are supported by mobile applications: GeoBilbao (reporting real-time information about parking and traffic status) and iBilbobus (information about bus stops and schedules) [66]. In Seoul, the Gangnam ubiquitous district has a centralized control framework, including the Transport Operation and Information Service (TOPIS) app and the "Owl Bus", which is an innovative bus service performing analytics on the big data that is collected on-board [82].

### 3.4. Smart Economy

A smart economy is based on the innovative interconnection of local and global markets through ICT, providing e-business and e-commerce services to increase productivity and delivery [83]. In addition, the concept of a sharing economy is also included in this domain, where individuals or private companies offer services exploiting their own assets as well as through peer-to-peer marketplaces. There are also peer-to-peer labor services,

in which citizens and stakeholders offer their work and experience for specific tasks [84]. Artificial intelligence and machine learning techniques have been implemented for building predictive models and improving recommendation systems for e-commerce and retail shopping [85]. The use of NFC and wireless sensor technologies has facilitated payment and transaction processes. In Shenzen, the use of mobiles and smartphones in daily transactions and information access is making cash and bankcards obsolete [86].

### 3.5. Smart Industry and Production

Smart industry and industry 4.0 define a transformation process in which IoT technologies, cyber-physical systems (CPS), M2M communication systems and cloud-based manufacturing [87] allow an innovative and less human-dependent productive environment [88]. Regarding the automation of goods supply chains, they can be easily tracked from the manufacturing process to final distribution using sensor technologies, such as RFID and NFC. Real-time information can be collected and analyzed for shipment tracking, as well as for the assessment of the quality and usability of products [14]. The smart industry and production domain includes all fields in which ICT leads to the automatization of the productive workflow, therefore also includes smart agriculture and farming, which addresses the challenge of sustainable food production. Smart agriculture systems often employ IoT devices to improve irrigation efficiency [89] and AI solutions are often deployed in IoT for agriculture, e.g., for crop monitoring, disease detection and data-driven crop supply management [7].

Dublin Airport has employed smart industry 4.0 solutions to replace its baggage-handling systems in Terminal 2 [90]. In Shenzen, a smart industrial complex includes a series of smart factories that share technologies and provide efficient services (exploiting an ultra-high-speed Internet connection, next-generation wireless network, free Wi-Fi infrastructures and IoT devices connected via a common cloud platform) [3].

### 3.6. Smart Energy

Smart energy systems involve the intelligent integration of decentralized renewable and sustainable energy sources and their efficient distribution [91] and aim to optimize power consumption [92]. Smart grids take advantage of ICT and IoT technologies for the better management of power generation and distribution, exploiting, for instance, prediction models (developed from collected consumption data) and often ensuring the self-healing of the energy network supply [93]. Smart grids help energy load balancing on the basis of usage and availability. In this way, it is possible to switch automatically to alternative sources of energy, as well as predicting future energy demand and estimating the power availability and price [14].

New generations of smart energy IoT devices have been designed for alternative energy harvesting, such as triboelectric nanogenerators (TENG) and electrostatic energy harvesters (EEH) [59]. A multiplicity of IoT sensors is involved in the smart energy context, such as light dependent resistors (LDRs), sensors for measuring light luminosity [94] and the consumption of solar radiation and electricity [95].

Nice, France, has made efforts to improve smart energy management by scheduling electricity consumption in residential and business locations. The smart grid in Nice was created through a smart solar neighborhood in city areas by supplying and storing distributed electricity [58]. The city of Padova has performed several initiatives in intelligent energy management, such as a smart lighting system in which each smart light device is geolocated in the city and is powered by photometer sensors that monitor the intensity of the light emitted by the lamps and check that the correct operation of the bulbs is performed [96]. Atlanta has implemented a smart neighborhood project to reduce the Home Energy Rating System (HERS) score. Energy optimization platforms manage home appliances, switching to solar power and batteries if available [3]. In Helsinki, smart grids help to reduce energy usage by 15% [97]. The Masdar City project aimed to be one of the most efficient and environment-friendly systems in the UAE by exploiting renewable

energy technologies. The city has planned a solar power array and rooftop solar panels that can provide more than 10 MW. This, combined with wind energy harvesting technologies, can supply energy for a target population of about 40,000 citizens [98].

### 3.7. Smart Environment

The smart environment domain includes environmental data collection, monitoring and analysis for pollution reduction, water quality and supply monitoring and weather and climate events management [67]. In this regard, air quality monitoring is a crucial factor for tracking levels of air pollutants (e.g., $NO_x$, $O_3$, $CO_2$, $N_2O$, PM10, PM2.5, etc.), which represent a serious issue for human health (caused by transportation, heating and industrial emissions). Smart waste management is also included in this area since it has numerous impacts on the environment. Control policies for waste production are handled with smart waste bins that are installed with sensors and are capable of providing the real-time analysis of the capacity that is currently available [92].

As for smart water, sensing devices that are devoted to the assessment of water quantity and quality typically measure parameters such as pH, conductivity, turbidity, total dissolved solids, etc. [99]. Electromagnetic and ultrasonic sensors are employed to measure the pressure for water consumption rate analysis [100]. The application of WSNs for water quantity and quality monitoring systems has opened up a new generation of smart water monitoring systems, providing a more advanced context awareness and near real-time interaction [101].

Smart environment applications and services are typically based on ambient and chemical sensors, which are used to measure physical quantities expressing environmental parameters and conditions, such as temperature, humidity, pressure [7] and different kinds of pollutants. Smart sensing and visualization technologies (satellites, LiDAR) are applied to greenhouse gas emissions (GHG) and land usage [99]. Location-based services and GIS data are also employed.

In Singapore, solid waste is managed by the Integrated Waste Management Facility (IWMF), which includes innovative IoT technology that allows for an increase in the process efficiency and a reduction in GHG emissions [102]. In Amsterdam, the Green City Watch is a geo-spatial AI platform that monitors urban green infrastructures in near real time using AI algorithms and satellite images [103]. In Stockholm, solar-powered smart waste bins are installed, which automatically report when they are fully charged and also perform waste packaging [66]. The Busan smart city, South Korea, employs smart water management systems in the whole urban water cycle [3]. In the context of the European TRAFAIR project, six European cities (Florence, Pisa, Livorno and Modena in Italy and Santiago de Compostela and Zaragoza in Spain) have adopted the Snap4City platform for monitoring urban air quality (using sensors that collect data in the six cities) and providing urban air quality predictions using simulation models [104].

### 3.8. Smart Healthcare

IoT technologies and ubiquitous computing have been widely applied to mobile healthcare for remote monitoring, telemedicine and telenursing, adverse drugs reactions, community healthcare, etc., and these aspects are even more relevant in this recent period of the COVID-19 pandemic. Remote patient monitoring (RPM) can be performed through the use of wearable or implanted devices (e.g., cardiac devices, airflow monitors, blood glucometers, etc.) that are connected in the cloud using WSN technologies [20]. This has led to the development of body sensor networks (BSNs) or wireless body area networks (WBANs), in which the integration of multiple heterogeneous data sources allows the acquisition of the biometric and physiological data of the patients for IoT healthcare applications [105]. Smart hospitals also rely on IoT technologies to provide services for medical staff and patients (the identification and monitoring of patients in hospitals and the smart management of medical instruments supporting decision-making processes in hospitals) [106]. All of these application fields and the related requirements impose well-defined standards, such

as Health Level Seven (HL7), PACS-DICOM in biomedical image processing [107], etc. Recent advances propose the use of AI techniques to define innovative applications, e.g., machine learning prognostics and the measurement of biometric parameters or symptoms from multimedia (using mobile videos or voice messages, through deep learning- and speech recognition-based methods) [108] and disease prediction and prevention [109].

Singapore has developed the HealthHub platform, which integrates personal health record management and the clinical data of patients and citizens [3]. In Stockholm, the New Karolinska Solna Hospital integrated smart energy and BIM technologies to provide user-focused services (for patients, visitors and medical staff) [110]. The hospital of Hefei, China, is one of the first of China's smart hospitals [111] where all aspects of the patients are managed via IoT and connected healthcare, also providing smart building services and sustainable energy management. The Helsinki University Hospital has implemented a real-time locating system (RTLS) to collect and share anonymized location data about on-site movements for proximity tracing during the COVID-19 pandemic. Moreover, cloud services are provided to enable doctors and nurses to remotely interact with COVID-19 patients [112].

To conclude the review, Table 1 presents an integrated summary of the smart city domains that were presented in this section, in terms of supported services and features, real-world case studies and the IoT technologies employed.

**Table 1.** A summary of smart city domains: services and features, IoT technologies employed and real-world cases.

| Smart City Domains | Services, Applications and Features | IoT and Sensing Technologies Involved | Real-World Cases |
|---|---|---|---|
| Smart Governance | - e-government<br>- Citizens' participation<br>- Collaborative and shared decision-making policies | - Web- and mobile-based applications for G2C, G2B and G2G [41,42] | - Barcelona [46,47]<br>- Amsterdam [48]<br>- Rio de Janeiro [3,49]<br>- Singapore [50]<br>- Toronto [51,52]<br>- Songdo [53,54] |
| Smart Living and Infrastructure | - Smart buildings<br>- Smart homes<br>- Smart tourism<br>- Smart education | - BIM models [57]<br>- Indoor/outdoor sensors and WSN technologies [59,62]<br>- Web and mobile apps, location-aware services, virtual and augmented reality and social media [64]<br>- E-learning systems and decentralized education [66] | - Los Angeles [68]<br>- Dubai [69] |
| Smart Mobility and Transportation | - Traffic management<br>- Dynamic routing<br>- Smart parking<br>- Vehicle sharing<br>- Sustainable mobility | - City sensors and actuators, personal devices<br>- IoT and intelligent transportation systems (ITSs) [71]<br>- Vehicle-to-vehicle communication (V2V), vehicle-to-infrastructure communication (V2I)<br>- VANETs [73]<br>- On-board diagnostic (OBD) tools [78] | - Berlin [79]<br>- Florence [3]<br>- Atlanta [3]<br>- London [80]<br>- Santander [81]<br>- Bilbao [66]<br>- Seoul [82] |
| Smart Economy | - e-business<br>- e-commerce<br>- Peer-to-peer marketplaces<br>- Peer-to-peer labor services | - AI solutions for web/mobile recommendation systems [85] | - Shenzen [86] |

**Table 1.** *Cont.*

| Smart City Domains | Services, Applications and Features | IoT and Sensing Technologies Involved | Real-World Cases |
|---|---|---|---|
| Smart Industry and Production | - Industry 4.0<br>- Smart manufacturing<br>- Predictive maintenance<br>- Smart agriculture and farming | - Cyber-physical systems (CPS)<br>- Cloud-based manufacturing [87] | - Dublin [90]<br>- Shenzen [3] |
| Smart Energy | - Energy management<br>- Sustainable energy harvesting<br>- Smart lighting<br>- Smart grids | - Triboelectric nanogenerators (TENG) and electrostatic energy harvesters (EEH) [59]<br>- Light luminosity sensors [94]<br>- Energy consumption measuring devices [95] | - Nice [58]<br>- Padova [96]<br>- Atlanta [3]<br>- Helsinki [97]<br>- Masdar City [98] |
| Smart Environment | - Air quality monitoring<br>- Weather monitoring<br>- Smart waste management<br>- Smart water | - Ambient sensors [7]<br>- Big data from satellite, LiDAR and GIS data [99] | - Singapore [102]<br>- Amsterdam [103]<br>- Stockholm [66]<br>- Busan [3]<br>- Florence, Pisa, Livorno and Modena, Santiago de Compostela and Zaragoza [104] |
| Smart Healthcare | - Telemedicine<br>- Remote patient monitoring (RPM) and healthcare tracking<br>- Smart hospitals<br>- e-health records management<br>- Disease prediction and prevention | - Wearable or implanted devices for remote patient monitoring (RPM) [20]<br>- Body sensor networks (BSNs) and wireless body area networks (WBANs) [105]<br>- Standards for information management (HL7, PACS-DICOM, etc.) [107] | - Singapore [3]<br>- Stockholm [110]<br>- Hefei [11]<br>- Helsinki [112] |

## 4. Discussion on Recent Trends, Open Challenges and Future Directions

IoT smart city technologies and applications are spreading rapidly, and this is reported in more and more real-world cases. However, on the basis of the analysis and review performed in the previous sections, this integration process is not complete since it is still facing some open challenges, which may be resolved in future developments.

For instance, there are interoperability problems due to the presence of many different IoT protocols, formats and frameworks [3,29,92,105,113], and this aspect is enhanced by the fact that many smart city applications have been initially developed as vertical silos applications [3,18] with each of them using its own solutions for data ingestion, storage and exploitation. Resolving the interoperability issues could bring economic benefits. In fact, achieving a higher level of interoperability among devices, applications and services involves the reduction in costs for producing completely new and different deployments of the solutions [113], thereby allowing backward compatibility through the exploitation of older systems as well as an incremental deployment and integration. On the other hand, the development of the IoT/IoE paradigm has led to the adoption of event-driven and push protocols [32], which has paved the way not only to sense the city but also to act through actuators and to create event-driven applications. However, most of the solutions proposed in the literature still focus on limited domains, addressing specific problems with little or no software reuse [3]. In order to handle the high variety of IoT devices and applications, the paradigm of microservice-oriented architecture is increasingly adopted in recent IoT-based solutions for smart cities [13,29,32,109]. This enhances the scalability and availability of IoT frameworks and simplifies the complexity of the traditional service-oriented architectures (SOA) [32]. For IoT-based solutions, the achievement of a higher scalability represents the possibility to efficiently collect and process increasingly larger amounts of data, which often leads to a higher accuracy in data analysis and often enables real-time or near real-time processing [114]. These aspects

imply important social involvement, for instance, in security and resilience, since they allow the building of more resilient tools that are able to perform real-time analysis and simulations, e.g., those used by local public authorities for early warnings and alerts in critical events, such as disaster management [99] and resilience planning. Moreover, many IoT solutions are still oriented toward traditional programming environments, while the most recent trends show an advancement in the exploitation of visual paradigm languages (VPL) [15,32], including Node-RED, for implementing workflows for IoT applications.

The adoption of microservice-oriented architectures leads toward the overcoming of the monolithic platform approach [3] since the microservice paradigm allows the reuse of software components and blocks. Moreover, microservice architectures are open to extensions and they can also exploit external services more easily (when necessary for delegating part of the computation as well as accessing additional services or applications) [29]. For instance, they can be easily adapted to support almost all kinds of IoT and communication protocols and also support data-driven and event-driven push modalities. This is moving in the direction of the implementation of smarter frameworks for business intelligence and data analytics, as well as more interactive visual analysis tools [32]. This deeper level of integration and complexity of IoT-enabled smart city platforms should bring advancements in performing real-time simulations, what-if analysis and supporting decision-making processes, which is the basis of the production of smarter and more efficient services and applications for all involved stakeholders.

Furthermore, IoT-enabled smart city platforms are evolving toward cross-organization and multitenancy IoT platforms and applications. This allows the development of large infrastructures that can support multiple organizations, enhance scalability and reduce the infrastructures' costs since they are shared between multiple operators [36]. This aspect is closely connected with the reuse of components in smart city frameworks and tries to harmonize and overcome the efforts needed in building custom-made platforms for each city or each specific context, which is economically inefficient [3]. Other potential future directions include the introduction and dissemination of novel network technologies, such as 5G [28,59,72,102]. Technological advances in networks and device solutions are important. In fact, the adoption of the most recent network technologies, such as 5G, combined with higher efficiency in building techniques and technologies following the paradigm of net zero energy infrastructure can lead to building solutions that are aiming for net zero carbon emissions [102]. Furthermore, the introduction of innovative computing paradigms, such as the introduction and integration of deep learning and AI solutions [109,115], semantic technologies and natural language processing (NLP) would improve the interaction level between smart devices and all smart city actors, as well as enabling the production of smarter services for a better quality of life.

Finally, in order to also discuss the societal challenges addressed by IoT and smart city technologies, we focused on assessing each application area and domain, as discussed in Section 3, in terms of their contribution to the SDGs. To this end, we will briefly introduce the 17 indicators representing the SDGs: (1) no poverty; (2) zero hunger; (3) good health and well-being; (4) quality education; (5) gender equality; (6) clean water and sanitation; (7) affordable and clean energy; (8) decent work and economic growth; (9) industry, innovation and infrastructure; (10) reduced inequalities; (11) sustainable cities and communities; (12) responsible consumption and production; (13) climate action; (14) life below water; (15) life on land; (16) peace, justice and strong institutions; and (17) partnerships for the goals. On the basis of our findings while reviewing the literature, relevant contributions and relations among the presented IoT–smart city application areas and the following SDGs were retrieved:

- Zero Hunger: Smart agriculture [7] solutions contribute to improving efficiency in accessing fundamental resources, such as food, and also allow precision agriculture [89];
- Good Health and Well-being: Smart healthcare solutions [106,108] contribute to improving efficiency in healthcare services that are provided in hospitals and medical structures, as well as at home. Big data collection and analysis in healthcare contexts

can be useful for monitoring critical cases, conditions and events [109], especially in the period of COVID-19 pandemic;

- Quality Education: Smart education solutions contribute to creating innovative education services, as well as to enhancing the interaction between remote and real-world learning activities [66];
- Clean Water and Sanitation: Smart water solutions [100] are employed to monitor the quantity and quality of water distribution and aim to minimize consumption and manage wastewater treatments [101]. This represents an important step in the proper design and maintenance of quality water systems;
- Affordable and Clean Energy: Smart energy solutions and energy grids [3,58,93,96,97] contribute to a more efficient energy distribution and usage [92], helping to minimize power consumption and consider innovative sustainable energy sources [91];
- Decent Work and Economic Growth: Smart governance solutions [3,38,39,46–54] contribute to economic growth [38] since they are expected to provoke a strong push in the direction of smart and digital public administrations [39]. Moreover, smart economy solutions [83,84] can also contribute to allowing citizens, companies and smart city stakeholders to follow the market for smart applications and data economy, rethinking the flexibility of jobs and labors [84] and, thus, redefining the economic value associated with them;
- Industry, Innovation and Infrastructure: Smart industry solutions [3,14,33,90] are establishing new and relevant digital infrastructures for sustainable industrial production [88] and data economy;
- Sustainable Cities and Communities: Several IoT-enabled smart city components contribute to improving the sustainability of smart city communities. For instance, smart mobility solutions [3,66,79–82] are aiming to establish near-to-zero emissions and reduced traffic flows and to also enhance the adoption of smart transportation and IoT paradigms. These aspects will bring relevant influences and improvements for the quality of life in smart cities [92];
- Climate Action: Smart environment technologies [3,66,102,103] that are focused on monitoring air quality and pollutant levels [99,104] contribute to analyzing and controlling air quality and fossil combustion, as well as their environmental impact in terms of $CO_2$, $NO$, $NO_2$, etc. (which are the main effects of fossil combustion);
- Peace, Justice and Strong Institutions: Smart governance solutions [3,38,39,46–54] contribute to providing institutions with data-driven decision-making processes [39], which makes citizens' participation more inclusive and deliberative, thus creating a consensus for the public good and enhancing equality and social justice [38].

## 5. Conclusions

In this paper, a review of the recent research literature on IoT-enabled smart cities framework was performed. The rationale behind this study was the requirement to understand and classify the most recent trends in the adoption of IoT technologies as a key driver for the efficient and sustainable development of smart cities. The purpose was also to highlight the main open challenges that need to be addressed and resolved in the future. The review was conducted both for key IoT technologies, which were analyzed following an architectural perspective, and for smart city approaches and frameworks, which were based on classification into eight domains describing the main application areas. From this analysis, it emerged that in recent years, the integration of IoT solutions and smart city frameworks is achieving increasingly higher levels of complexity and wider application ranges, which go beyond the past generation of vertical silo applications that were based on specific domains. In fact, the necessity to overcome vertical silos (i.e., where data are collected and "siloed" in a unique system and closed to the rest of IoT [114]) has provoked several initiatives. For instance, the EU developed specific initiatives and IoT programs, such as the open API standard Open Messaging Interface (O-MI) and Open Data Format (O-DF) [116]. This produces new added-value across multiple platforms, providing the

possibility to share protocols, data and results and allowing the better and more efficient cooperation between all involved actors and stakeholders (users, software and network providers, institutions, companies, etc.) [114]. To this end, the new generation of smart applications will manage and optimize more complex sets of heterogeneous information, data, systems, sensors, devices, etc. However, this process is still not complete since it has to cope with several open technical and social challenges (regarding the efforts to harmonize the many different standards for IoT formats and protocols, interoperability and scalability issues and the achievement of sustainability goals) in order to move toward microservice-oriented architectures, event-driven/data-driven applications and more sustainable solutions. Finally, another important driver could be represented by all involved stakeholders and actors, who are raising their awareness and becoming more actively engaged in the smart city environment, not only as service consumers but also as the producers of valuable contents and information.

**Author Contributions:** Conceptualization and methodology, P.B., G.P. and P.N.; research and investigation, P.B., G.P. and P.N.; writing—original draft preparation, P.B., G.P. and P.N.; writing—review and editing, G.P.; funding acquisition, P.N. All authors have read and agreed to the published version of the manuscript.

**Funding:** The solution has been partially funded by the HERIT-DATA Interreg project.

**Acknowledgments:** The authors would like to thank the HERIT-DATA Interreg project. Snap4City (https://www.snap4city.org) is an open technology and research by DISIT Lab, University of Florence, Italy.

**Conflicts of Interest:** The authors declare no conflict of interest.

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
