# Peer review of "IoT-Enabled Smart Cities: A Review of Concepts, Frameworks and Key Technologies"

_applsci, doi:10.3390/app12031607_

Round 1
Reviewer 1 Report
This is a well written review article in an important and new field of innovation. Here are som feedback/reflections I would like to make:
- The first aim for the paper is to provide an updated overview of literature about Smart Cities domain as well as for IoT key technologies and applications. This part of the article is the strongest part.
- However, there are some changes I would like to see. In section 3.1 row 265 you propose that IoT technologies are transforming city governance on the basis of different actors. There is a lack of an explanation for each of the types what IoT will do for the relationship.
- In the same section I miss at least three smart city projects which are the Sidewalk lab in Toronto, the SongDo smart city in South Korea and the Masdar City project in UAE.
- It is very technology driven article - Where are the societal challenges all this technology will be developed for? Could you connect the different application areas to challenges that we have in society? For example the Sustainable Development Goals.
- The second aim is to give insight on recent trends, open challenges and future directions which is a bit vague. The answer to this aim lies in section 4 where you discuss different topics such as 1)Interopability problems, 2) Micro service oriented architecture which enhances scalability and availability 3) Cross Organization and Multi-Tenancy IoT plattforms and 4)applications and the novel network technologies. Could you better connect there new trends to what they can achieve for society? Why do we need to have this technology in place? What are the societal benefits and challenges with these new solutions?
- In the conclusion section- could you give some more information om what does it mean that Smart Cities are being more complex and also go beyond vertical silos and that more stakeholders can be involved? Are there benefits or drawbacks?
Reviewer 2 Report
The paper introduces a review about IoT technologies used in Smart Cities. Firstly, the authors introduce the main IoT technologies used in Smart Cities and then they present a series of use cases distributed in 8 conceptual areas. The field of research is relevant. However, I have some comments to the authors before to consider this review to publication.
- Many papers have been published about this topic, including reviews, as the authors state in some parts of the paper. Proof of that is that the authors have based this paper on “52 surveys and reviews on IoT-enabled Smart Cities” as they state. Thus, what does this review add to what has already been published in the literature? The contribution is not clear.
- The five layers used in section section 2 to review IoT technologies are based just on two papers. A review article must be supported by a broad consensus of the scientific literature.
- Important technologies are missing in section 2.2 (e.g. NB-IoT, LTE-M, Z-Wave, WIMAX, among others)
- Authors state that there are messaging systems to allow persistence and to exchange data in IoT network and they provide some examples: Orion Context Broker (from FIWARE), Apache Kafka and RabbitMQ. While these services provide an easy way to collect and exchange information, they do not provide database-level persistency as they are just data brokers. As an example, Orion Context Broker, just stores the last message received by a node in a non-relational databased based on MongoDB. To allow data persistence, FIWARE provides another service called CYGNUS which allows to connect Orion to different databases (MySQL, PostgreSQL, CKAN, Apache Hadoop, etc.). Thus, that affirmation must be corrected and contextualized.
- In section 2.4, application layer, authors state that “Event-driven IoT applications constitute the Business Intelligence of an IoT-enabled platform”. This statement is too categorical (moreover it is not supported by any reference).
- In the same section as before, authors state that these event-driven platforms are “based on Visual Programming Languages (VPL)”. Again, another categorical statement which it is not supported by anything. In addition, authors introduce some of these platforms: NETLab, Ardublocks, Scratch and Node-RED. These platforms, in my opinion, are not tools for professional use (they are mainly used by hobbists and by elementary school students) specially in the case of the first three. Moreover, these platforms are used to program low-end devices (microcontrollers), with the exception of Node-RED (scratch can be used both for microcontrollers and applications). Thus, these platforms should not be in the application layer but in the lower layers (sensing).
- The rest of the paper is a review of different use cases of IoT technologies over Smart Cities which, as said before, are extracted from other reviews as the author state, followed by a discussion. In my opinion, the discussion is one of the most important parts in reviews, since is the part in which the authors identify the current advances and lacks in the topic studied. The discussion of this review is vague and contain categorical statements not supported by the literature.
Round 2
Reviewer 1 Report
The article is improved a lot since my last review. I have nothing more to say and this it is ready to be published.
Reviewer 2 Report
Authors have addressed most of my comments and suggestions.
I have a last suggestion prior to the publication of the paper:
- Although I can understand that the authors want to give more value to the discussion by including the SDGs, I don't think it adds much to the relationship they make.